# Community perceptions towards invasion of *Prosopis juliflora*, utilization, and its control options in Afar region, Northeast Ethiopia

**Wakshum Shiferaw**[1]*, **Sebsebe Demissew**[2], **Tamrat Bekele**[2], **Ermias Aynekulu**[3]

**1** College of Agricultural Sciences, Natural Resources Management, Arba Minch University, Arba Minch, Ethiopia, **2** College of Natural and Computational Sciences, Addis Ababa University, The National Herbarium, Addis Ababa, Ethiopia, **3** World Agroforestry Centre (ICRAF), Nairobi, Kenya

* waaqsh@yahoo.com

**Data Availability Statement:** All relevant data are within the manuscript and its Supporting information files.

## Abstract

This study aimed to assess community perceptions towards invasion of *Prosopis juliflora*, utilization, and its control options in Afar region, Northern Ethiopia. Using purposive sampling and stratified random methods, 20 members of key informants and 154 households from four sites of Awash Fentale and Amibara Districts were selected. For data analysis, we used Kruskal Wallis non-parametric tests of K independent samples. About 30% of respondents in Amibara and 29% in Awash Fentale reported that *Prosopis juliflora* was largely introduced into their landscape by livestock. It showed that 29% of the respondents in Awash Fentale and 41% in Amibara responded that *Prosopis juliflora* largely invaded and affected rangelands. Morevover, about 1% of respondents in Awash Fentale and 14% in Amibara argued that *Prosopis juliflora* hindered movements of livestock. In addition, 30% of respondents in Amibara and 29% in Awash Fentale believe that *Prosopis juliflora* was largely dispersed by livestock. It showed that 20% of households in Awash Fentale and 41% in Amibara have the notion that *Prosopis juliflora* majorly impacted rangelands. Whereas 1.3% of respondents in Awash Fentale and 14% in Amibara argued that *Prosopis juliflora* have hampered the movement of livestock. Thus, the afromentioned findings are implications for management of rangelands. With regard to the control of *Prosopis juliflora* invasions, 12% of respondents in Awash Fentale and 33% in Amibara District tried control its expansion by fire. About 10% of respondents in Awash Fentale and 9% in Amibara district managed *Prosopis juliflora* expansion by its utilization, whereas, in Awash Fentale (11%) and Amibara (8%) households indicated that invasion of *Prosopis juliflora* could be controlled by mechanical methods. It is advisable to do some managerial work to reverse these impacts as perceived by local communities in the study area to avert the aggressive proliferation of *Prosopis juliflora* in the region.

**Funding:** The authors received no specific funding for this work.

**Competing interests:** The authors have declared that no competing interests exist.

## Introduction

*Prosopis juliflora* (Sw.) DC. (hereafter *P. juliflora*) is a shrub or tree species native to Mexico, Central, and Northern America. From its native ranges, *P. juliflora* spread to Africa, Asia, and Austria [1]. In Africa, *P. juliflora* was first introduced in Senegal in 1822 and continued to establish in other countries at different times [2]. In Ethiopia, it was first introduced in the 1970s to restore degraded lands [3]. After its introduction, *P. juliflora* started to aggressively expand. It has been become an invasive or noxious weed in several African countries including Kenya, Ethiopia, Sudan, Senegal, and South Africa [1,7]. In the introduced countries, *P. juliflora* expansion increased. For example, PENHA [4] reported that the global cover of *P. juliflora* as a whole was 50 million hectares and *P. juliflora* covered about 5 million hectares in Africa in 2014 while Shiferaw et al. [5] reported a land cover of $1.20 \times 10^6$ ha by *P. juliflora*. It was encroaching at a rate of $3.11 \times 10^4$ ha yr$^{-1}$ and constituted 12.3% of the land surface in the Afar region. Pittroff [6] also reported that *P. juliflora* cover was more than $1.80 \times 10^6$ ha of Afar region. It has now become an invasive or noxious weed in several African countries including Kenya, Ethiopia, Sudan, Senegal, and South Africa [1,7].

Thus, community perception will play a significant role in rangeland management and lay the conceptual foundation for the management of the invasion by *P. juliflora*. Study by Dafalla [8] indicates that educated people are more supportive of the eradication of *P. juliflora* than people that are not educated. This is debatable! Less educated people are to some degree dependent only on wood of *P. juliflora* for livelihood support. In contrary, in Ethiopia, local people have negative attitudes towards *P. juliflora*. They believe that the species has replaced economically important pasture and farmlands, and threatening pastoral and agro-pastoral livelihoods. It is also thought to have impacted human and animal health. The species is has become a threat to road traffic, and water infrastructures. In addition, the invasion has turned into a major driver of biodiversity loss in the invaded regions [3].

Inadequate management practices like prevention of its invasion into rangelands by local development interventions and social conflicts over grazing lands facilitated the invasion of *P. juliflora* in Afar region [9–11]. The perceptions of Afar pastoralists concerning the P. juliflora invasion are negative because it impacts their livelihoods and environment they inhabit [12]. Likewise, the majority of the households in the Gewane district of Afar region have not appreciated the positive and significant association of *P. juliflora* with their income diversifications [13]. Palatable grasses including *Chrysopogon plumulosus* Hochst., *Cenchrus ciliaris* L., *Setaria verticillata* (L.), and other valuable woody species such as *Acacia tortilis* (Frossk.), *A. senegal* (L.) Willd., *A. nilotica* (L.) Willd. ex. Del. was being replaced by inavsaion of the species. Thus, the present study aimed to assess community perceptions towards (i) the intoruciton and invasion (ii) the socio-economic values of the species, and (iii) controlling options of *P. juliflora* in Afar region of Ethiopia.

## Materials and methods

### Description of the study area

Amibara District is located in between altitudes of 741 and 746 m.a.s.l. It is located between 9° 19′ 44′′ N and 40° 10′ 52′′ E, whereas Awash Fentale is located at 700 and 1000 m.a.s.l. and 9° 10′ 00′′ N and 40° 03′ 33′′ E.

The mean annual temperature for the Awash Fentale District was 27 ± 2°C, while the mean minimum was 16.7 ± 1.97°C. The mean maximum temperature was 37.8 ± 2.1°C (Fig 1a). The mean annual temperature for Amibara District was 26.8 ± 4°C, whereas the mean minimum temperature was 13.8 ± 4.3°C and the mean maximum was 38.2 ± 2.3°C (Fig 1). The study

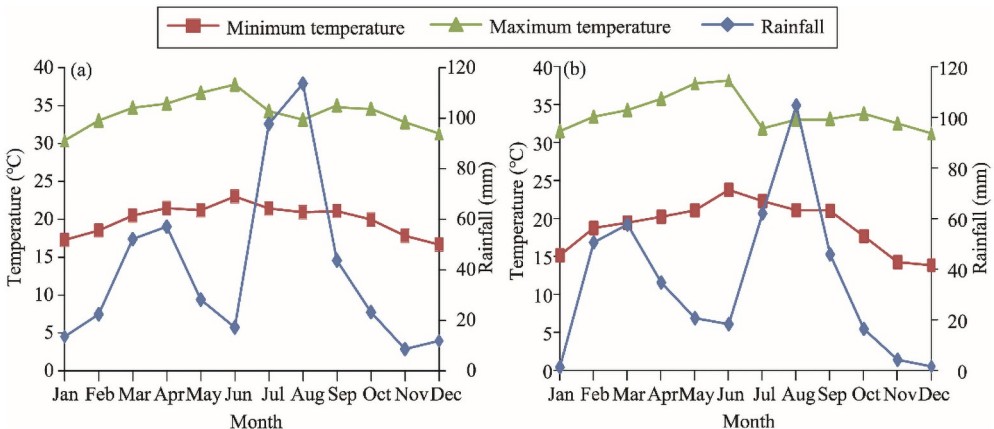

**Fig 1.** (a) Thirty-one year climate diagram for Awash Fentale District and (b) Fifteen year climate diagram for Amibara District [16].

areas are located within semiarid and arid agro-ecologies of Ethiopia. The annual precipitation of Awash Fentale and Amibara districts was 490 ± 34 mm and 416 ± 31 mm respectively (Fig 1). A total population of 83, 851 and 40,901 was living in Amibara and Awash Fentale respectively [14]. Ninety percent of Afar people are pastoralists, while another 10% are considered as agro-pastoralists [15].

Afar region is characterized by desert and semi-desert scrubland, *Acacia-Commiphora* woodland, and bush land vegetation types [17]. This study was conducted in *Acacia-Commiphora* woodland and desert and semi-desert scrubland with vegetation subtype *Acacia-Commiphora* woodland and bushland. The characteristic herbaceous vegetation consisted of *Chrysopogon*, *Sporobolus*, *Dactyloctenium*, *cymbopogon*, and *Cynodon* species. The woody vegetation was mainly composed of *Acacia Senegal* (L.) Willd., *Acacia oerfota* (Forssk.) Schweinf., *Acacia nilotica* (L.) Willd. ex. Del., *Acacia tortilis* (Frossk.) Hayne, *Acacia mellifera* (Vahl) Benth., *Acalypha acrogyna* Pax, *Cadaba rotundifolia* Forssk., *Dobera glabra* (Forssk.) Poir., *Grewia tenax* (Forssk.) Fiori, *Salvadora persica* L., *Balanites aegyptiaca* (L.) Del., and *Ziziphus spina-christi* L. [17].

## Data collection

**Socio-demographicof household characteristics.** We used semistructured and structured questionnaires to collect through key informants interview and household survey. Primary data were being collected through discussion with key informants and sampled households using pre-tested semistructured and structured questionnaires. For data collection, a three stage sampling method i.e., districts were purposely selected while household respondents were randomly selected. We took random households for the household from two sites from each district namely Diduba and Kebena from Awash Fentale and Kurkura and Andido from Amibara disricts [18]. The respondents for households were randomly selected females and males that were heads of the family. The memebers of the key informants were selected from district experts that were related to natural resources management experts, team leaders of district natural resources manament, office leaders of each district agricultural and natural resources, and adiminstrators of each district.

Diduba and Kurkura sites were less affected by *P. juliflora* compared to Kebena and Andido. A toral of 154 household which is 5% of total households was selected for household

survey from the four sites. The selected households were from lightly, moderately, and highly invaded sites. Sample households from total households in the study sites were then stratified into wealth, sex and, age categories and then selected using simple random sampling technique from the total households in the sites.

Based on the information from key informants households which had $> 10$ camels, $> 20$ cattle and $> 60$ small ruminants (goats and sheep) were categorized as rich. Those owning 1–10 camels, 5–10 cattle, and 10–60 small ruminants were categorized as medium households while hoseholds with no camels, $< 5$ cattle and $< 10$ small ruminants were categorized as poor households [19].

## Data analysis

Qualitative and quantitative methods were used during the data analyses. The data were not normally distributed, thus non-parametric tests of K independent samples of Kruskal Wallis of $\chi^2$ for mean separtion was used. The empirical multinomial logit model for this study was specified as [20]:

$$yi = f(x1, x2 \ldots xn)$$

Where yi, the dependent variable, is the wealth or socioeconomic status of pastoralists, xi's are the included explanatory variables. The dependent variable (yi) is defined as follows: 1 for the poor pastoralists, 2 for the middle-class pastoralists, and 3 for the rich pastoralists. yi is also defined as 1 for male and 2 for female pastoralists; 1 as the youth (1–18 years), 2 for adult class (19–60 years), and 3 for old class ($> 65$ years) of pastoralists. Aside from the wealth status, other variables initially considered for inclusion in the model include age, sex, and education status of the agropastoralists and pastoralists. Then, explanatory vs. response variables were used for the empirical valuations, all the analyses were done using the XX and YY procedures of descriptive statstics in SPSS Software [21].

Households were significantly different among their opinions regarding questions raised such as: what were their mode of living, how *P. juliflora* was introduced, why it was introduced into their sites, preferred site for *P. juliflora* regeneration, benefits they get from *P.juliflora*, use of *P. juliflora* for traditional medicine, the preparation of *P. juliflora* for traditional medicine for human disease, livestock disease, preparation method for traditional medicine for livestock diseases from *P. juliflora (P < 0.05)*. However, the rests of the households' perceptions didn't show significant (*P > 0.05*) (S2 Table).

The variable definitions and measurements are given in S1 Table. For assessing community awareness towards the invasion of *P. juliflora*, 32% from Awash Fentale and 66.2% households from Amibara District were used for the interview. Among the respondents, 71% and 29% were male and female households, respectively (Table 1). Of which 1%, 91%, and 8% of households fall in young, middle, and elder age groups, respectively. The family size of households ranged from 0–13 and mean value of 6. About 95% of the respondents had no formal education, 1% had primary educations, and 2% attended secondary and post-secondary education. Among the respondents, 27% were representatives of peasant associations and also had different positions in government offices.

## Results

### History of *P. juliflora* introduction

We found that about 30% of the respondent in Amibara and 29% in Awash Fentale districts thought that *P. juliflora* was introduced to Afar by livestock. About 19% of the respondents in

**Table 1. Household characteristics of Awash Fenatle and Amibara Districts.**

| Explanatory variable | Mean of household characteristics | SE | Minimum | Maximum | *df* | $\chi^2$ |
|---|---|---|---|---|---|---|
| District | 1.7 | 223.2 | 1 | 2 | 1 | 16.23 |
| Site | 2.8 | 154.0 | 1 | 4 | 3 | 19.25 |
| Sex | 1.3 | 226.5 | 1 | 2 | 1 | 24.96 |
| Age | 2.1 | 285.2 | 1 | 3 | 2 | 230.7 |
| Wealth | 1.4 | 211.0 | 1 | 3 | 2 | 83.08 |
| Relationship to household head | 1.8 | 116.7 | 1 | 8 | 6 | 525.36 |
| Household type | 1.6 | 142.6 | 1 | 6 | 5 | 354.75 |
| Household number | 5.7 | 93.9 | 0 | 13 | 13 | 79.27 |
| Education of household head | 0.1 | 216.9 | 0 | 3 | 3 | 400.23 |
| Education of household member | 0.8 | 163.8 | 0 | 3 | 5 | 161.66 |

Amibara and 3% Awash Fentale districts thought argued that *P. juliflora* was introduced to by a foreigner came to Amibara area. A small proportion of the respondents in Amibara (6.5%) and Awash Fentale (1.3%) thought that the species was introduced by local people (Table 2). About 22% of key inormatns in Amibara and 33% in Awash Fentale thought that the species was introduced by a foreigner (Fig 2).

## Purposes of *P. juliflora* introduction

About 31% of the respondents in Amibra and 10% in Awash Fentale reported that fuelwood was the main reason for the introduction of *P. juliflora* in Afar region (Fig 2). Households in Amibara (18%) and Awash Fentale (5%) reported that the species was introduced for shade purpose (Fig 2). About 7% of the respondents in Amibara and 9% in Awash Fentale reported that the species was introduced for the purpose of soil and water conservation (Fig 2). However, 12% of the respondents in Amibra and 17% in Awas Fentale districts did not know the purpose of its introduction.

According to the key inforamts, 11% in Amibara and 22% in Awash Fentale confirmed that *P. juliflora* was introduced for fuelwood purposes. About 33% of the key informants in Amibara and 22% in Awash Fentale reported that the species was introduced for shade purpose (Fig 3).

**Table 2. Agents for the introduction of *P. juliflora* into Awash Fentale and Amibara Didtricts.**

| Response | | Frequency | % |
|---|---|---|---|
| Local people | Awash Fentale | 2 | 1.3 |
| | Amibara | 10 | 7 |
| Natural | Awash Fentale | 0 | 0.0 |
| | Amibara | 5 | 3 |
| Foreigners | Awash Fentale | 5 | 3 |
| | Amibara | 29 | 19 |
| Livestock | Awash Fentale | 44 | 29 |
| | Amibara | 46 | 30 |
| Wild animals | Awash Fentale | 0 | 0.0 |
| | Amibara | 1 | 0.6 |
| Others | Awash Fentale | 0 | 0.0 |
| | Amibara | 11 | 7 |

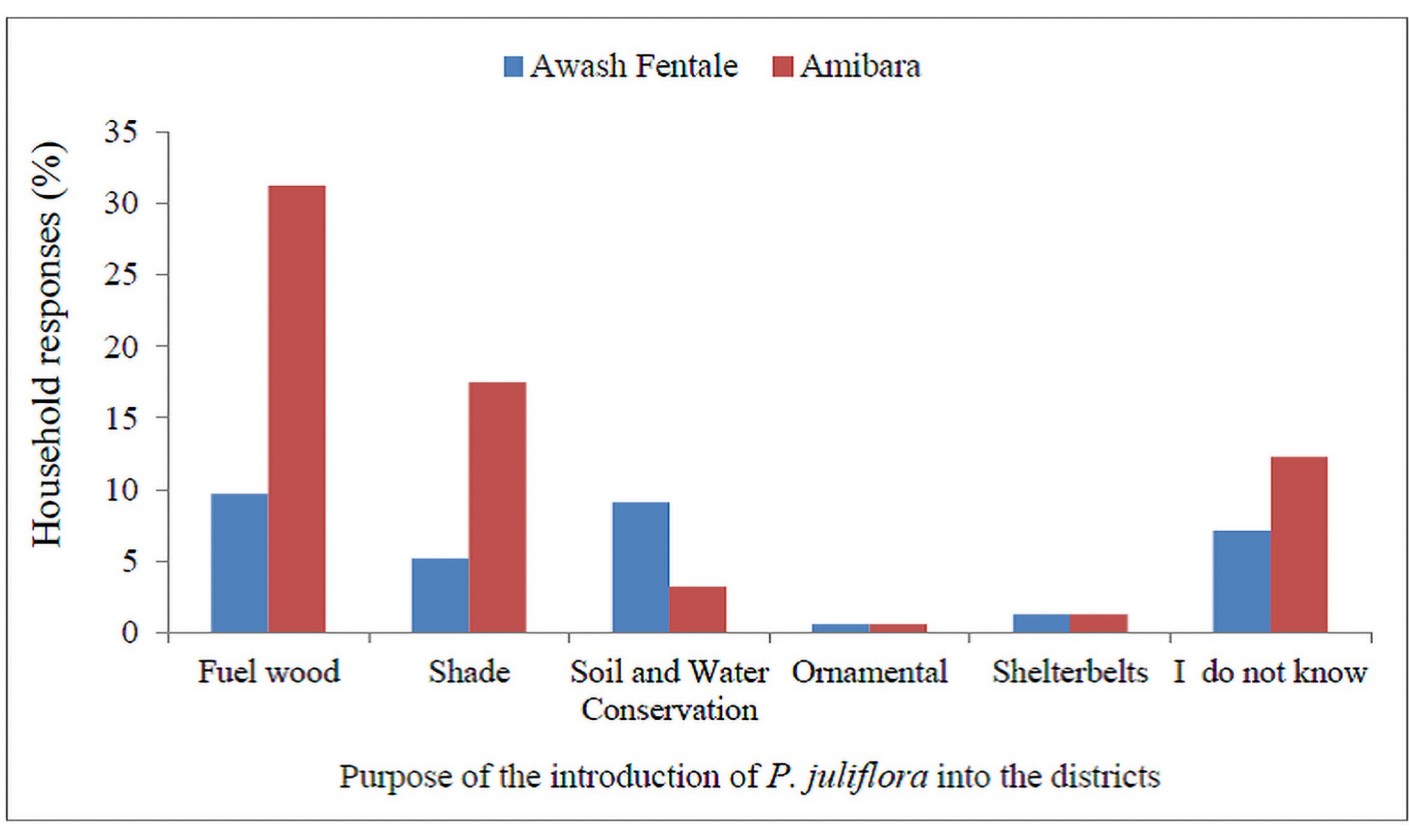

**Fig 2. Household responses on the reasons for the introduction of *P. juliflora* into Awash Fenatle and Amibara Districts.**

## Socio-economic values of *P. juliflora*

Regarding the opinions of households, the use of *P. juliflora* for traditional medicine, preparations of human traditional medicine from plant parts, human disease cured by *P. juliflora*, livestock diseases cured by *P. juliflora*, and preparations of livestock traditional medicine from plant parts had shown that significant variations among respondents ($P < 0.05$) across Districts (S2 Table). Of households, about 49% of the respondents in Amibara and 12% in Awash Fentale district reported that they got different benefits from *P. juliflora* (Table 3).

## Use of *P. julflora* for medicinal purposes

About 10% of households in Awash Fentale and 9% in Amibara District used *P. juliflora* for traditional medicines. It was reported that 8% of respondents in Awash Fentale and 5% in Amibara District used *P. juliflora* for curing human injury that pricked by its thorns. Only 8% and 4% of respondents in Awash Fentale and Amibara District perceived that livestock injures were cured by traditional medicine of *P. juliflora* (Table 4).

## Preparation of traditional medicine from *P. juliflora*

For preparation of traditional medicine curing human diseases (wounds), 8% and 5% of households in Awash Fentale and Amibara responded that it cured by crushing its leaf part. But, the majority of households in Awash Fentale (25%) and Amibara (59%) didn't know how to prepare traditional medicines for human diseases, whereas 8% and 4% of households in

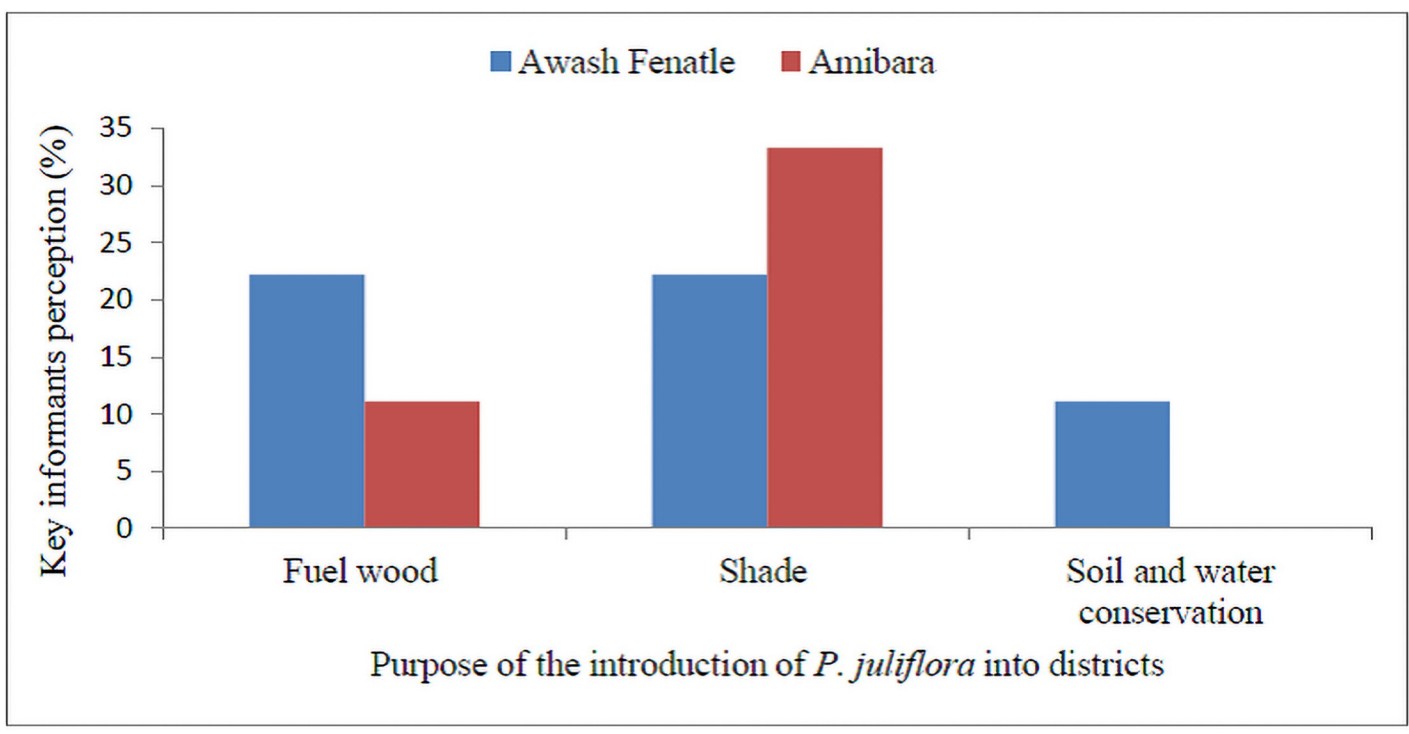

**Fig 3. Key informants responses for purposes of introduction of *P. juliflora* into Awash Fenatle and Amibara Districts.**

Awash Fentale and Amibara responded that livestock wounds cured by pounding its leaf part. But, the majority of households in Awash Fentale (26%) and Amibara (62%) didn't know how to prepare its traditional medicines for curing livestock (Table 4).

## Effects of *P. juliflora*

Larger number of repondents in Amibara (41%) and Awash Fentale (20%) reported that *P. juliflora* affected them most by invading and degrading the health of their rangelands, whereas a few hoseholds in the study districts reported that *P. juliflora* blocked roads e.g., human and livestock roads (Table 5).

## Effect of *P. juliflora* invasion and land use changes

About 16% of households in Awash Fenatle and 27% in Amibara District showed that conversions of land use land cover were caused by invasion of *P. juliflora*, whereas 11% in Awash Fenatle District and 16% in Amibara were caused by farm land expansion. In addition, 0.6% of respondents in Awash Fentale and 6% in Amibara Distrcits reported that the cause for conversions of land uses were drought or shortage of rainfall (Table 6).

Households responded that 13% in Awash Fenatle and 35% in Amibara District changed from woodlands into *P. juliflora*, whereas 16% in Awash Fentale and 27% in Amibara district changed from grazing lands into *P. juliflora* cover. Moreover, 3% in Awash Fentale and 27% households in Amibara District reveald that water bodies changed into the *P. juliflora* cover (Fig 4).

**Table 3. Perceptions of households on benefits of *P. juliflora* in Awash Fenatle and Amibara Districts.**

| Benefits | District | Frequency | % |
|---|---|---|---|
| Shade for livestock | Awash Fentale | 9 | 5.8 |
| | Amibara | 11 | 7.1 |
| Shade for people | Awash Fentale | 2 | 1.3 |
| | Amibara | 4 | 2.6 |
| Fuel wood | Awash Fentale | 10 | 6.5 |
| | Amibara | 66 | 42.9 |
| Furniture | Awash Fentale | 1 | 0.6 |
| | Amibara | 0 | 0 |
| House construction | Awash Fentale | 3 | 1.9 |
| | Amibara | 6 | 3.9 |
| Live fence | Awash Fentale | 10 | 6.5 |
| | Amibara | 2 | 1.3 |
| Soil and Water Conservation | Awash Fentale | 1 | 0.6 |
| | Amibara | 1 | 0.6 |
| Ameliorating effects | Awash Fentale | 0 | 0 |
| | Amibara | 1 | 0.6 |
| Shelterbelt | Awash Fentale | 0 | 0 |
| | Amibara | 1 | 0.6 |
| Fodder | Awash Fentale | 2 | 1.3 |
| | Amibara | 0 | 0 |
| Combating desertification | Awash Fentale | 5 | 3.2 |
| | Amibara | 1 | 0.6 |
| Medicinal values | Awash Fentale | 9 | 5.8 |
| | Amibara | 10 | 6.5 |

## Effects of *P. juliflora* invasion and its control

Of the hoseholds, 3% in Awash Fentale and 6% in Amibara had the attitudes that *P. juliflora* formed impenetrable thicket which hindered the easy movements of human beings around. Results also show that 11% of key informants in Awash Fentale and 22% in Amibara District argued that prime grazing lands of Afar region were invaded by *P. juliflora* (Fig 5). It was argued that 55% of households in Amibara and 27% in Awash Fenatle Districts reported that *P. juliflora* had negative impacts on biodiversity. However, the rest 11% in Amibara and 7% in Awash Fentale Districts had positive impacts on biodiversity. In addition, the majority of households (45%) indicated *P. juliflora* affected livestock in Amibara District and 27% in Awash Fentale District (Table 7).

As a result, 33% of key informants in Awash Fentale and 22% in Amibara confirmed that livestock production and productivity reduced. In Awash Fentale and Amibara District, 7% and 11% of households agreed that *P. juliflora* impacted biodiversity. About 44% and 22% of key informants in Awash Fentale and Amibara also confirmed as an invasion of *P. juliflora* had negative impacts on biodiversity in terms of the reduction of species diversity and the rest 11% and 22% of key informants in Awash Fentale and Amibara perceived that *P. juliflora* didn't affect species diversity (Fig 5).

About 27% of households in Awash Fentale and 45% in Amibara District responded that livestock were the most affected by *P. juliflora* invasions. In addition, 7% and 20% of respondents in Awash Fentale and Amibara responded that plants affected by *P. juliflora* invasions. 21% and 38% of household respondents in Awash Fentale and Amibara District responded

**Table 4. The use of *P. juliflora* for traditional medicine in Awash Fenatle and Amibara Districts.**

| Do you think *P. juliflora* used for traditional medicine? | | | |
|---|---|---|---|
| Response | District | Frequency | % |
| Yes | Awash Fentale | 14 | 10 |
| | Amibara | 14 | 9 |
| No | Awash Fentale | 34 | 22 |
| | Amibara | 85 | 55 |
| I do not know | Awash Fentale | 4 | 3 |
| | Amibara | 3 | 2 |
| Which part of *P. juliflora* used for traditional medicine? | | | |
| Seeds or pods | Awash Fentale | 14 | 9 |
| | Amibara | 14 | 9 |
| Leaf | Awash Fentale | 38 | 25 |
| | Amibara | 88 | 57 |
| Preparation of *P. juliflora* for traditional medicine for human cure? | | | |
| Pounding | Awash Fentale | 13 | 8 |
| | Amibara | 11 | 7 |
| I don't know | Awash Fentale | 39 | 25 |
| | Amibara | 91 | 59 |
| Human diseases cured by traditional medicine of *P. juliflora*? | | | |
| Wound | Awash Fentale | 13 | 8 |
| | Amibara | 8 | 5 |
| I do not know | Awash Fentale | 39 | 25 |
| | Amibara | 93 | 60 |
| Livestock diseases cured by traditional medicine of *P. juliflora*? | | | |
| Wound | Awash Fentale | 12 | 8 |
| | Amibara | 6 | 4 |
| I don't know | Awash Fentale | 40 | 26 |
| | Amibara | 95 | 62 |
| Preparation method for traditional medicine for livestock diseases from *P. juliflora*? | | | |
| Pounding | Awash Fentale | 12 | 8 |
| | Amibara | 7 | 5 |
| I don't know | Awash Fentale | 40 | 26 |
| | Amibara | 94 | 61 |

**Table 5. Impacts of *P. juliflora* in Awash Fenatle and Amibara Districts.**

| Effects | Districts | Frequency | % |
|---|---|---|---|
| Woody weedy in agricultural lands | Awash Fentale | 12 | 8 |
| | Amibara | 7 | 5 |
| Invasion into grazing lands | Awash Fentale | 30 | 20 |
| | Amibara | 63 | 41 |
| Blocking roads of livestock | Awash Fentale | 2 | 1.3 |
| | Amibara | 22 | 14 |
| Blocking roads of human beings | Awash Fentale | 2 | 1.3 |
| | Amibara | 9 | 6 |
| Invasion of water courses, drying rivers and water tables | Awash Fentale | 0 | 0 |
| | Amibara | 1 | 0.6 |
| Lack of aesthetic value due to its monoculture | Awash Fentale | 5 | 1.3 |
| | Amibara | 0 | 0 |

**Table 6. The causes of land use land cover conversions in Awash Fentale and Amibara Districts.**

| Causes | District | Frequency | % |
|---|---|---|---|
| Livestock production beyond the carrying capacity/overgrazing | Awash Fentale | 17 | 11 |
| | Amibara | 25 | 16 |
| Farm land expansion | Awash Fentale | 4 | 3 |
| | Amibara | 0 | 0.0 |
| Anthropogenic | Awash Fentale | 0 | 0.0 |
| | Amibara | 1 | 0.6 |
| Invasion of *P. juliflora* | Awash Fentale | 24 | 16 |
| | Amibara | 42 | 27 |
| Moisture stress | Awash Fentale | 2 | 1.3 |
| | Amibara | 0 | 0.0 |
| Drought or shortage of rainfall | Awash Fentale | 1 | 0.6 |
| | Amibara | 9 | 6 |
| Toxic effects of *P.juliflora* | Awash Fentale | 2 | 1.3 |
| | Amibara | 24 | 16 |

that *P. juliflora* had impacts on plants. About 12% hoseholds (in Awash Fentale) and 33% (in Amibara) had attitudes that livestock were among those affected by the invasion of *P. juliflora* (Table 8).

In Awash Fentale (10%) and Amibara Districts (9%) households had forwarded their suggestions that invasion of *P. juliflora* could overcome through management by utilization; whereas, in Awash Fentale (12%) and Amibara (33%) households indicated that invasion of *P. juliflora* could be controlled by fire. The other 11% households in Awash Fenatle and 8% in Amibara District reported that invasion of *P. juliflora* could be controlled by mechamical methods such as: mechanical control by cutting mature trees, mechanical control by uprooting

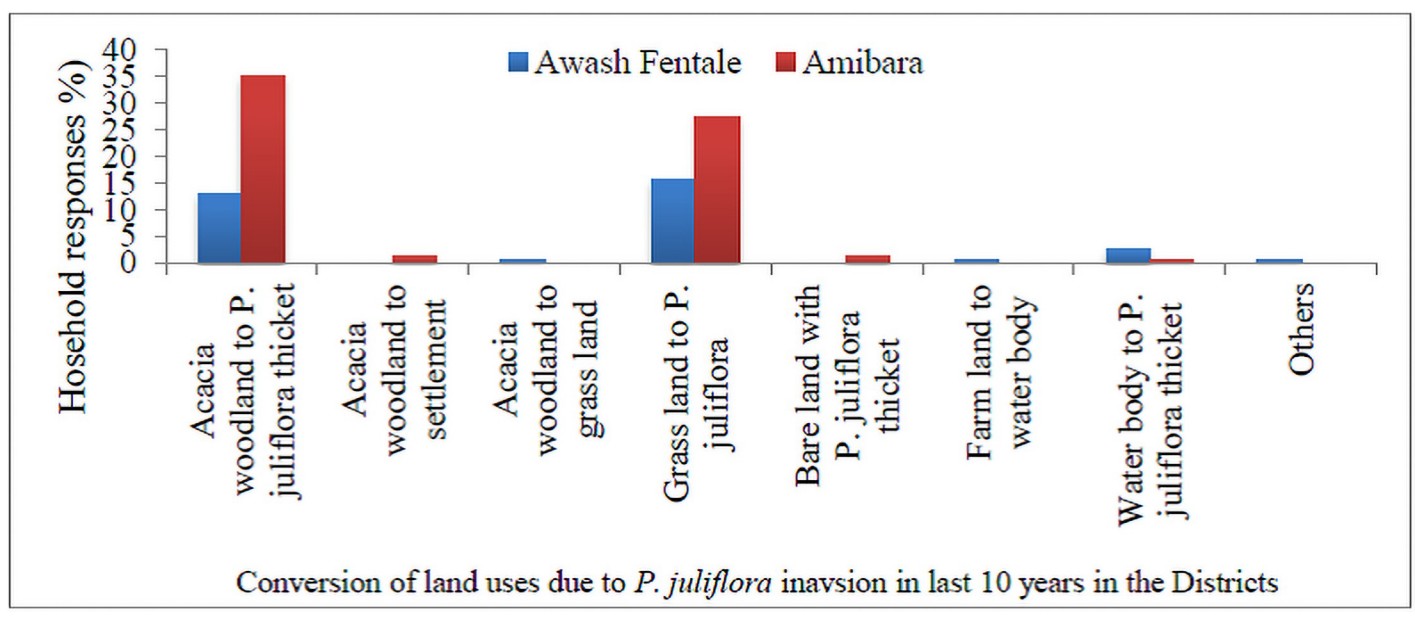

**Fig 4. Conversion of land uses due to *P. juliflora* inavsion in last 10 years in Awash Fentale and Amibara Districts.**

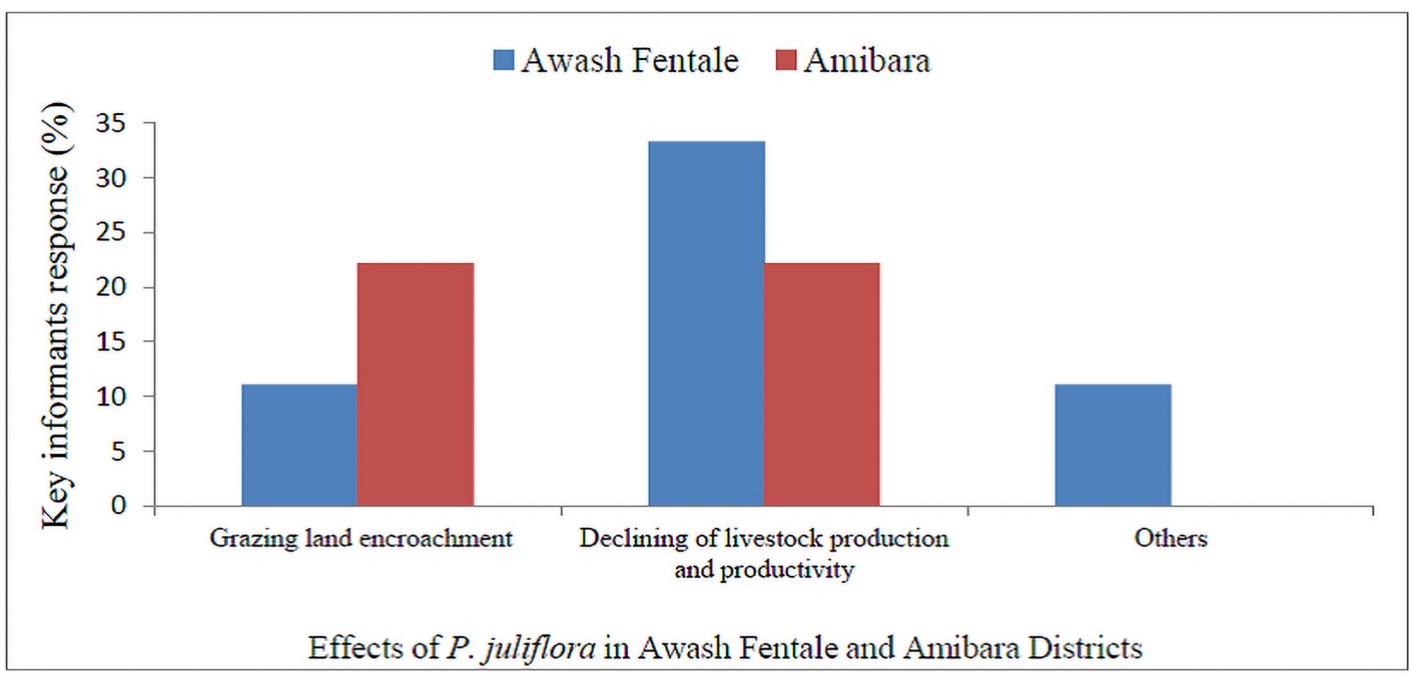

**Fig 5. Responses of key informants about impacts of *P. juliflora* in Awash Fenatle and Amibara Districts.**

mature trees, mechanical control by cutting juvenile stems, and mechanical control by uprooting juvenile stems (Fig 6).

About 22% key informants (in Awash Fentale) and 44% (in Amibara) had attitudes that livestock were among those affected by the invasion of *P. juliflora*. Furthermore, in Awash

**Table 7. Types of impacts on biodiversity in Awash Fenatle and Amibara Districts.**

| What kinds of impacts biodiversity on biodiversity? | | | |
|---|---|---|---|
| | District | Frequency | % |
| Positive | Awash Fentale | 10 | 7 |
| | Amibara | 17 | 11 |
| Negative | Awash Fentale | 42 | 27 |
| | Amibara | 85 | 55 |
| What type of biodiversity affected by *P. juliflora*? | | | |
| Livestock | Awash Fentale | 42 | 27 |
| | Amibara | 69 | 45 |
| Plants | Awash Fentale | 10 | 7 |
| | Amibara | 31 | 20 |
| Human beings | Awash Fentale | 0 | 0 |
| | Amibara | 2 | 1.3 |
| Which will be the most affected biodiversity by *P. juliflora*? | | | |
| Livestock | Awash Fentale | 47 | 31 |
| | Amibara | 69 | 45 |
| Plants | Awash Fentale | 5 | 3 |
| | Amibara | 29 | 19 |
| Human beings | Awash Fentale | 0 | 0 |
| | Amibara | 1 | 0.6 |
| Wild animals | Awash Fentale | 0 | 0 |
| | Amibara | 3 | 2 |

**Table 8. Perceptions of key informants on impacts of *P. juliflora* on biodiversity in Awash Fenatle and Amibara Districts.**

| What kinds of impacts of *P. juliflora* on biodiversity? | | | |
| --- | --- | --- | --- |
| Response | District | Frequency | % |
| Positive | Awash Fentale | 1 | 11 |
| | Amibara | 2 | 22 |
| Negative | Awash Fentale | 4 | 44 |
| | Amibara | 2 | 22 |
| What type of biodiversity affected by *P. juliflora*? | | | |
| Livestock | Awash Fentale | 2 | 22 |
| | Amibara | 4 | 44 |
| Human beings | Awash Fentale | 1 | 11 |
| | Amibara | 0 | 0 |
| Plants | Awash Fentale | 2 | 22 |
| | Amibara | 0 | 0 |
| What type of biodiversity most affected by *P. juliflora*? | | | |
| Livestock | Awash Fentale | 3 | 33 |
| | Amibara | 4 | 44 |
| Plants | Awash Fentale | 2 | 22 |
| | Amibara | 1 | 11 |

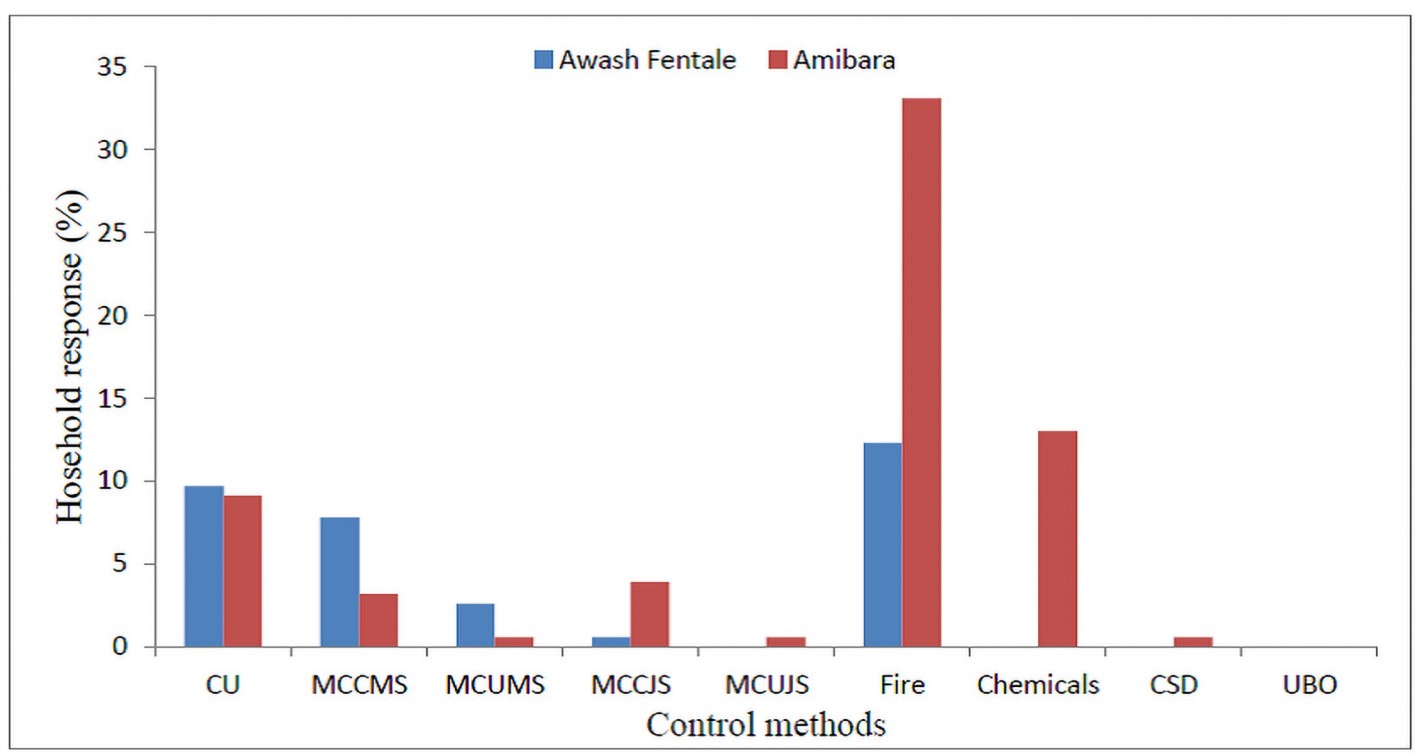

**Fig 6. Responses of households about the control of invasion of *P. juliflora* in Awash Fentale and Amibara Districts.** Note: CU = Manage by utilization, MCCMS = Mechanical control by cutting mature stems, MCUMS = Mechanical control by uprooting mature stems, MCCJS = Mechanical control by cutting juvenile stems, MCUJS = Mechanical control by uprooting juvenile stems.

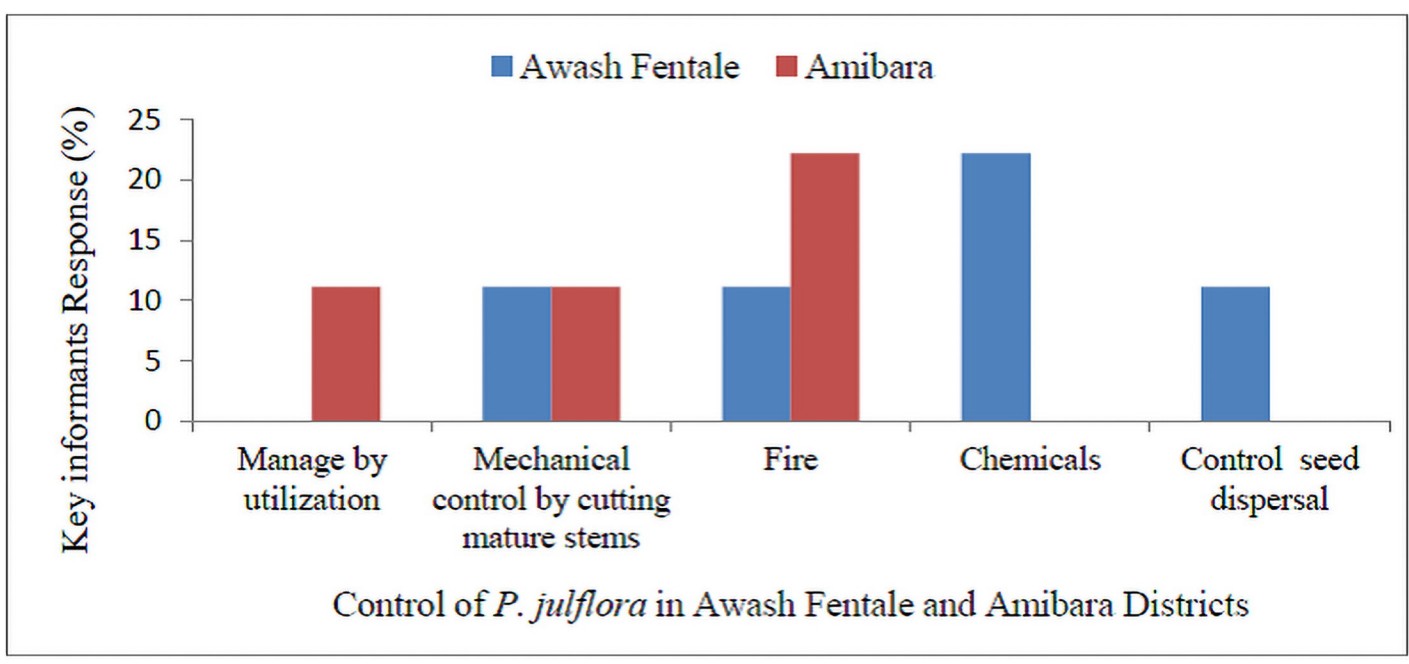

**Fig 7. Responses of key informants about the control of invasion of *P. juliflora* in Awash Fentale and Amibara Districts.**

Fentale (22%) and Amibara (11%) key informants indicated that invasion of *P. juliflora* reduced plant species diversity. In Awash Fentale (56%) and Amibara districts (44%) key informants had forwarded their suggestions that invasion of *P. juliflora* could overcome through management by utilization. Unless the invasion of the species can be controlled in the future, in the Awash Fentale (22%) and in Amibara District (44%) key informants informed that prime grazing lands will be overtaken by its invasion (Fig 7).

## Discussion

### Lifestyle of households

The majority of households in Awash Fentale were agro-pastoralists than those in Amibara District. The reason could be due to the existence of wetlands around Kebena River in Kebena site of Awash Fentale that most households engaged in farming and livestock rearing activities. Similarly, Tegegn [22] indicated that most households in the Gewane district of Ethiopia were agro-pastoral ways of living.

Few households in the study areas engaged in non-farm activities and obtained their incomes from daily labor wages and petty trades shopping in small scale. Similar findings by Shackleton [23] state that households engaged in non-farm activities obtained their incomes in other engagements including employment in government institutions. Moreover, in the arid Kalahari of South Africa, Shackleton [24] argued that few households engaged in non-farm activities and earned their incomes from daily labor activities.

### Households' perceptions to the introduction history of *P. juliflora*

In the study sites, the reasons for the introduction of *P. juliflora* varied among households and key informants. The reasons could be due to little knowledge on who introduced *P. juliflora* and its means introduction into a new inavsion [25]. Reports also show that the introduction of *P. juliflora* was not clear in Afar region in particular and Africa in general [26].

Similar findings by [27] reveal that introduction of most exotic plant species were related to lifestyle changes of communities and contradictions in views and perceptions. Mwangi and Swallow [28] argued that purposes of introduction of *P. juliflora* into most countries of Africa remain unclear and the local people were not engaged in its introduction. Contrary to the local communities' perceptions, research reports [29] and [30] indicate that *P. juliflora* was introduced by workers in Middle Awash Irrigation Project in Afar region of Ethiopia.

## Households' perceptions towards the uses of *P. juiflora*

The majority of key informants in both District s stated that introduction of *P. juliflora* was for purpose of shade. Some scholars such as [26] and [30] suggested more general views indicating the introduction was to combat desertification. According to the findings of this study, *P. juliflora* is used for shade purposes either for livestock or human beings [24,26,30–32]. On the other hand, a study by [12] shows that *P. juliflora* was used as livestock feed.

## Use of *P. julflora* for medicinal purposes

Unlike the present study, research report by Shackleton [33] reveals that farmers' used *P. juliflora* for shade purposes greater than used for fuelwood and medicine. However, findings by Shackleton [34] show that the use of *P. juliflora* for medicinal purposes is greater that for purposes of construction and land rehabilitation. On the other hand, Shackleton [33] reported that the use of *P. juliflora* for medicine was greater the uses of *P. juliflora* for fodder, fuelwood, and shade purposes, but less than that of pods for food of children and sloving job opportunities. In contrast to the present findings, Shackleton [34] reported that most households used *P. juliflora* for medicinal purposes. Likewise, studies by Argaw [35] and Ibrahim [36] argue that *P. juliflora* used as versatile medicine and important and a potential candidate for deriving phytomedicines.

According to Preeti [36]) and Henciya [37], the curative powers of traditional medicines for human beings or livestock could be due to high contents of extracts such as flavonoids, tannins, alkaloids, quinones, or phenolic compounds that demonstrate potentials in various antimicrobial activities such as analgesic, anthelmintic, and antibiotic constituents. In Ethiopia, few literatures had reported on the parts of *P. juliflora* that were used for traditional medicine, on the methods of its preparation for human beings or livestock diseases, or about its curing effect. Walter [38] also reported on the use of the nectar and pollen of *P. juliflora* to produce honey. The author further argued that *P. juliflora* didn't rank equally with those of indigenous species that had been known by people for thousands of years for their medical powers in India.

A study by Wise [39] shows from the Northern Cape of South Africa that pod of *P. juliflora* had medicinal properties and is being used for medicinal purposes. In addition, research report by Shackleton [32] reveals that pods used to produce organic medicine were being gathered from local traders of cities, towns, and villages across South Africa's Northern Cape. Besides, Pasiecznik [1] reported that all parts of *P. juliflora* tree had uses of traditional medicines. They further reported on the occasional medicinal use of *P. juliflora* by Native Americans to treat the epidemic diseases of Old World Origin.

## Effects of *P. juliflora*

The majority of households reported that invasion of *P. juliflora* invaded more into rangelands in Amibara than Awash Fentale District. The reasons could be due to the initial introduction of the species into Amibara District. Similarly, Harnet [40] reported that the invasion of *P. juliflora* into both dry season and wet season rangelands and roadsides in the Eastern lowlands of Eritrea. A study by Argaw [34] also shows that the overall analysis of responses showed rangelands were among the impacts of *P. juliflora*. In addition, Ndhlovu [41] also suggested that

wider areas of rangelands covered by *P. juliflora* invasions and reduced its grazing capacity by 34% in South Africa.

Similar research reports confirmed that intermingling growth and thorny natures of *P. juliflora* could be due to the formation of thicket that blocked access roads for animals or human beings [34,42–44]. Invasion of *P. juliflora* into rangelands made blocked the roadsides and resulted in the losses of prime grazing lands. Thus, the production and productivity of livestock in the *P. juliflora* invaded areas were being reduced. Several researchers such as [45–47] and [5] also confirmed that the invasion of *P. juliflora* resulted in the loss of prime grazing lands and ultimately the causes for the reduction of livestock production and productivity.

Overall, less positive attitudes were reflected by households about the impacts of *P. juliflora* than negative impacts, and this might be due to a lack of full scale awareness about the impacts of *P. juliflora*. Moreover, most of the respondents lost their livestock that the health of animals by *P. juliflora* and few of them used *P. juliflora* for income sources. These types of views were long-standing issues of hot debates in numerous other countries in the tropics and sub-tropics. A similar view in Sudan shows that the benefits from *P. juliflora* completely outweigh its detrimental effects in this particular area [48]. The individuals in the tropics and subtropics accepted *P. juliflora* as positive where native tree species were not growing in their area.

Studies have shown that people's views to be shaped by the negative or positive attributes of the invasive plant species [25]. Respondents who had large number of livestock were likely to be more aware of the negative impacts of *P. juliflora* than the positive effects [12]. On the contrary, several types of research show that *P. juliflora* has negative impacts on plant diversity [39,40,49,50]. On the other hand, Maundu [51] argued that *P. juliflora* was an aggressive weed with both strong positive and negative attributes. The negative perceptions towards *P. juliflora* slowly changed back to positive attitudes for the tree after communities recognized on how to exploit the tree for their economic and social benefits [52].

The variations in the views of the stakeholders on the impacts of *P. juliflora* on biodiversity might be due to the difference in their knowledge and practices regarding *P. julidlora*. Most scholars argued that *P. juliflora* affected biodiversity [26,34,39,50]. Most researchers reported that the invasion of *P. juliflora* influenced mostly plants [53–56].

Large proportions of households in Awash Fentale District were of the opinion that human beings were being injured by the long thorns of *P. juliflora* [34,48,56], whereas higher proportions of households in Amibara District show that impenetrable thicket of *P. juliflora* blocked roadsides that hindered the easy movement of both humans and animals [11,22,28,34]. The variations in the responses were due to the difference in the severity of invasions of *P. julflora* and the level of knowledge level between households in the two Districts.

A report by Seid [12] shows that burning of *P.juliflora* was the most widely employed method to control the invasion of *P. juliflora*. Control of *P. juliflora* by burning was in line with the arguments of most key informants in the Amibara District. Most key informants in Awash Fentale on the other hand, tried to control *P. juliflora* invasion using chemical methods. The study by [33] reveals that control of *P. juliflora* by utilization that is charcoal making, fuelwood, and contruction purpose was the best method to minimize the invasion *P. juliflora*. In the future, if the expansion of *P. juliflora* is not controlled, most households will predict that it will largely invade and affect prime grazing lands [33].

## Factors shaping knowledges and perceptions of households

The living modes of the communities were also likely to affect invasion of *P. juliflora*. Most of the time and in all places of Afar region, the communities might also hear the negative sides of *P. juliflora* than positive by extension agents and experts in their sites.

The most palatable part of *P. juliflora* by animals was leaf of *P. juliflora* and the most toxic plant part killing their animals after their consumptions [1]. The reasons could be due to a permanent weakening of the ability to digest cellulose in pods. This might also be due to the high sugar content of the pod that depressed the rumen bacterial cellulose activity and finally killing the animal [43,56]. In this study, the decline in the number of livestock feeds could be due to the decline in the capacity of rangelands for livestock grazing through suppressing and displacing important indigenous forage species. Similar results were reported by [3,28,41,50] from their studies in Kenya, South Africa, and Ethiopia indicate the places or countries!. Most non-formal educated households in Awash Fenatle and primary school educated households in Amibara argued that *P. juliflora* was being introduced by NGOs and an individuals which similar to [3,18].

## Preferresd sites for invasion of *P. juliflora*

In Amibara and Awash Fentale District s, non-formal, primary, secondary, and tertiary-educated households ranked rangelands > homestead > mechanized farmlands > roadsides with respect to preferred sites for the establishment of *P. juliflora* respectively. Findings by [24,30,33] showed that rangelands were changed more into *P. juliflora* than other land-use types. The reason could be due to greater probability of seed dispersal by fecal droppings and abiotic suitability in the rangelands and in homesteads (e.g., moisture and high organic matter accumulation).

## Conclusion

The perceptions of local communities towards invasive species depend on the level of communities' awareness, knowledge, and practices in their socio-economic and ecological uses. The benefits of invasive species outweigh their negative effects. When this is the case, local communities tend to embrace species and retain them an ecosystem.

The majority of households in Awash Fentale and few households in Amibara District follow the agro-pastoral mode of living. The majority of the households specialized on on-farm activities or livestock production for their income sources. A few households in the study areas engaged in off farm activities e.g., daily laborers and petty trades. The majority of households viewed that livestock were the main causes for the spread of *P. juliflora* in their areas. Few people in Amibara district were of the opinion that a foreigner working in the Middle Awash Irrigation project was the culprit for the introduction of *P. juliflora* into the Afar region.

In general, the household use of *P. juliflora* is minimal in the study area. The majority of households were not using *P. juliflora* at all. Some households used *P. juliflora* as shade for livestock. A few households used *P.juliflora* for traditional medicines to cure their livestock and human beings. The leaf of *P. juliflora* was the part mainly used than other parts of the plant.

The severe invasion of rangelands was indicative of the fact that livestock production and productivity were ultimately hampered in the study area, thus making agro-pastoral livelihood at risk. *P. juliflora* also blocked roads and pathways so much that accessibility of grazing areas became more difficult for livestock and human beings. The majority of households in both Districts were of the view that *P. juliflora* had no impacts on biodiversity particularly on plants. Moreover, the species has encroached into lower topographic areas: homesteads, and wetland areas. People had made attempts to control *P. juliflora* using fire, mechanical clear-cutting, and control through utilization (e.g., charcoal making, utilization of its woods for different purposes). Thus, it's advisable in ussing *P. juliflora* to redue its further invasion into rangelands and other landuses in the region.

## Supporting information

**S1 Table. Variables definition and measurements [57].**
(DOCX)

**S2 Table. Effects of site variation on perceptions of communities towards introduction, use, and effects of *P. juliflora* in Amibara and Awash Fentale Woredas.**
(DOCX)

**S1 File.**
(DOCX)

## Author Contributions

**Conceptualization:** Wakshum Shiferaw, Ermias Aynekulu.

**Data curation:** Wakshum Shiferaw.

**Formal analysis:** Wakshum Shiferaw, Ermias Aynekulu.

**Funding acquisition:** Wakshum Shiferaw.

**Investigation:** Wakshum Shiferaw, Ermias Aynekulu.

**Methodology:** Wakshum Shiferaw, Ermias Aynekulu.

**Project administration:** Wakshum Shiferaw, Sebsebe Demissew, Tamrat Bekele.

**Resources:** Wakshum Shiferaw.

**Software:** Wakshum Shiferaw.

**Supervision:** Wakshum Shiferaw, Tamrat Bekele, Ermias Aynekulu.

**Validation:** Wakshum Shiferaw, Sebsebe Demissew, Tamrat Bekele, Ermias Aynekulu.

**Visualization:** Wakshum Shiferaw, Sebsebe Demissew, Tamrat Bekele.

**Writing – original draft:** Wakshum Shiferaw.

**Writing – review & editing:** Wakshum Shiferaw, Tamrat Bekele, Ermias Aynekulu.

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
