## [Editor Report · Decision Letter 0]

22 Oct 2021

PONE-D-21-18433Pastoralists’ perception towards the invasion of Prosopis juliflora in Afar region, Northeast EthiopiaPLOS ONE

Dear Dr. Gemeda,

Thank you for submitting your manuscript to PLOS ONE. After careful consideration, we feel that it has merit but does not fully meet PLOS ONE’s publication criteria as it currently stands. Therefore, we invite you to submit a revised version of the manuscript that addresses the points raised during the review process. Please submit your revised manuscript by December 6, 2021. If you will need more time than this to complete your revisions, please reply to this message or contact the journal office at plosone@plos.org. Please include the following items when submitting your revised manuscript:A rebuttal letter that responds to each point raised by the academic editor and reviewer(s). You should upload this letter as a separate file labeled 'Response to Reviewers'.A marked-up copy of your manuscript that highlights changes made to the original version. You should upload this as a separate file labeled 'Revised Manuscript with Track Changes'.An unmarked version of your revised paper without tracked changes. You should upload this as a separate file labeled 'Manuscript'.

We look forward to receiving your revised manuscript.

Kind regards,

Tunira Bhadauria, Ph.D.

Academic Editor

PLOS ONE

Additional Editor Comments (if provided):

I was reviewing the authors' amended work, and while I believe they did an excellent job of responding to the reviewers' criticisms and suggestions, I still believe the language should be improved significantly, as there are numerous grammatical problems. I think authors seek assistance from somebody who has an excellent grasp of the English language, rewrite the work as needed, then resubmit it for publication.

Journal Requirements:

2. Thank you for including your ethics statement: "We agreed to work our institution cooperation letter and study districts".   

Please amend your current ethics statement to confirm that your named institutional review board or ethics committee specifically approved this study.

3. Please include your tables as part of your main manuscript and remove the individual files. Please note that supplementary tables (should remain/ be uploaded) as separate "supporting information" files"

4. Please provide additional details regarding participant consent. In the ethics statement in the Methods and online submission information, please ensure that you have specified (1) whether consent was informed and (2) what type you obtained (for instance, written or verbal, and if verbal, how it was documented and witnessed). If your study included minors, state whether you obtained consent from parents or guardians. If the need for consent was waived by the ethics committee, please include this information.

5. Thank you for stating the following financial disclosure: "No Initials of the authors who received each award"

6. Thank you for stating the following in your Competing Interests section: "None of the authors have a conflict of interest"

7. In your Data Availability statement, you have not specified where the minimal data set underlying the results described in your manuscript can be found. PLOS defines a study's minimal data set as the underlying data used to reach the conclusions drawn in the manuscript and any additional data required to replicate the reported study findings in their entirety. All PLOS journals require that the minimal data set be made fully available. For more information about our data policy, please see http://journals.plos.org/plosone/s/data-availability.

8. We note that you have stated that you will provide repository information for your data at acceptance. Should your manuscript be accepted for publication, we will hold it until you provide the relevant accession numbers or DOIs necessary to access your data. If you wish to make changes to your Data Availability statement, please describe these changes in your cover letter and we will update your Data Availability statement to reflect the information you provide.

9. Your ethics statement should only appear in the Methods section of your manuscript. If your ethics statement is written in any section besides the Methods, please move it to the Methods section and delete it from any other section. Please ensure that your ethics statement is included in your manuscript, as the ethics statement entered into the online submission form will not be published alongside your manuscript. 

10. Please ensure that you refer to Figure 5 in your text as, if accepted, production will need this reference to link the reader to the figure.

11. We note that Figure 1 in your submission contain map images which may be copyrighted. All PLOS content is published under the Creative Commons Attribution License (CC BY 4.0), which means that the manuscript, images, and Supporting Information files will be freely available online, and any third party is permitted to access, download, copy, distribute, and use these materials in any way, even commercially, with proper attribution. For these reasons, we cannot publish previously copyrighted maps or satellite images created using proprietary data, such as Google software (Google Maps, Street View, and Earth). For more information, see our copyright guidelines: http://journals.plos.org/plosone/s/licenses-and-copyright.

12. Please include captions for your Supporting Information files at the end of your manuscript, and update any in-text citations to match accordingly. Please see our Supporting Information guidelines for more information: http://journals.plos.org/plosone/s/supporting-information. 
---

## [Author Response · Author response to Decision Letter 0]

8 Dec 2021

Answer

Journal: PLOS ONE

Manuscript ID: PONE-D-20-29179

Editor concerns: There are various serious concerns: i) the analysis conducted with the data: a serious pitfall in transforming categorical data before undertaking non-parametric analysis; the results, interpretation and conclusion are not therefore convincing; ii) lots of unnecessary details are given in the results and the discussion sections which clearly impede the focus of the message; iii) the manuscript needs a thorough editing to improve readability and grammatical oddities that are maintained throughout.

Answer: We tried to edit and reduce the errors; the manuscript is edited to correct grammatical errors and its readability. The manuscript is substantially revised and changed by the authors especially by corresponding author (Dr. Wakshum Shiferaw), Co-authors: Prof. Tamrat Bekele and Dr. Ermias Aynekulu. Prof. Sebsbe Demissew is another co-author who had given technical advisee for the improvement of the paper.

Reviewer #1

Concern: (1) You noted that the data were not normally distributed which is probably correct. Most of the input data are in fact categorical and should not be transformed. The data were then transformed before being analyzed using non-parametric methods. The transformation is inappropriate and may have affected your results. The data must be analyzed using methods designed for such data

Answer: The data now analyzed using methods designed for data analysis before transformation 

Concern (2): You only mention the SPSS package but not the actual routines/tools/methods used to do the analysis. It is essential that you are as explicit as possible in describing what you did because I need to know that you used an appropriate tool/method. The results and the discussion sections are very lengthy and loaded with detail which would be of little interest to most of the prospective readers (and is generally evident in the figures). Try to reduce them to the essentials and try to focus on exploring the reasons why people may have given the responses they did and why there were differences between the districts that you sampled. The more succinct you can be the more likely it is that people will read and cite your paper.

Answer

Now we tried to mention the appropriate methods used for the analysis

The length section of results and discussions are reduced

Reviewer #2

Concerns

1. It has to be acknowledged that the method used in this study need some clarification;

Answer: now it is clear 

2. The writing style in all sections of the paper need to be revised;

Revised 

3. The introduction must be restructured to clearly show and demonstrate the relevance and importance of your study as well as its added value to the wealth of knowledge on the same subject;

Answer: Improved

4. The data analysis is largely descriptive;

Yes the nature of the title

5. The section “results” is so long and not straightforward;

Answer: reduced 

Specific comments are well taken and the document is improved

---

## [Editor Report · Decision Letter 1]

13 Dec 2021

Community perceptions towards invasion of Prosopis juliflora, utilization , and its control options in Afar region, Northeast Ethiopia

PONE-D-21-18433R1

Dear Dr. Gemeda

We’re pleased to inform you that your manuscript has been judged scientifically suitable for publication and will be formally accepted for publication once it meets all outstanding technical requirements.

Kind regards,

Tunira Bhadauria, Ph.D.

Academic Editor

PLOS ONE

Additional Editor Comments (optional):

The paper has been thoroughly edited, with the authors taking into account the reviewers' suggestions as well as their remarks in order to improve the manuscript. I believe the work is scientifically appropriate for publication in the journal, and I recommend that it be accepted for publication.
---

## [Editor Report · Acceptance letter]

17 Jan 2022

PONE-D-21-18433R1 

Community perceptions towards invasion of *Prosopis juliflora*, utilization, and its control options in Afar region, Northeast Ethiopia 

Dear Dr. Shiferaw:

I'm pleased to inform you that your manuscript has been deemed suitable for publication in PLOS ONE. Congratulations! Your manuscript is now with our production department. 

Kind regards, 

on behalf of

Dr. Tunira Bhadauria 

Academic Editor

PLOS ONE